# Shape analysis of gamma rhythm supports a superlinear inhibitory regime in an inhibition-stabilized network

**R Krishnakumaran**[1], **Mohammed Raees**[2], **Supratim Ray**[1,2]*

**1** IISc Mathematics Initiative, Department of Mathematics, Indian Institute of Science, Bangalore, India,
**2** Centre for Neuroscience, Indian Institute of Science, Bangalore, India

* sray@iisc.ac.in

**Data Availability Statement:** The LFP Data and code for replication of our results are accessible at the Github repository: https://github.com/

## Abstract

Visual inspection of stimulus-induced gamma oscillations (30–70 Hz) often reveals a non-sinusoidal shape. Such distortions are a hallmark of non-linear systems and are also observed in mean-field models of gamma oscillations. A thorough characterization of the shape of the gamma cycle can therefore provide additional constraints on the operating regime of such models. However, the gamma waveform has not been quantitatively characterized, partially because the first harmonic of gamma, which arises because of the non-sinusoidal nature of the signal, is typically weak and gets masked due to a broadband increase in power related to spiking. To address this, we recorded local field potential (LFP) from the primary visual cortex (V1) of two awake female macaques while presenting full-field gratings or iso-luminant chromatic hues that produced huge gamma oscillations with prominent peaks at harmonic frequencies in the power spectra. We found that gamma and its first harmonic always maintained a specific phase relationship, resulting in a distinctive shape with a sharp trough and a shallow peak. Interestingly, a Wilson-Cowan (WC) model operating in an inhibition stabilized mode could replicate this shape, but only when the inhibitory population operated in the super-linear regime, as predicted recently. However, another recently developed model of gamma that operates in a linear regime driven by stochastic noise failed to produce salient harmonics or the observed shape. Our results impose additional constraints on models that generate gamma oscillations and their operating regimes.

## Author summary

Gamma rhythm is not sinusoidal. Understanding these distortions could provide clues about the cortical network that generates the rhythm. Here, we use harmonic phase analysis to describe these waveforms quantitatively and show that gamma rhythm recorded from the primary visual cortex of macaques has a signature arch shaped waveform, with a sharp trough and a shallow peak, when visual stimuli such as full-screen plain hues and achromatic gratings are presented. This arch shaped waveform is observed over a wide range of stimuli, despite the variation in power and frequency of the rhythm. We then compare two population rate models that have been used to accurately describe the

RKrishnakumaran/GammaHarmonicsProject-
HuesDataset.

**Funding:** This work was supported by Wellcome
Trust/DBT India Alliance (Senior fellowship IA/S/
18/2/504003 to SR) and DBT-IISc Partnership
Programme. The funders had no role in study
design, data collection and analysis, decision to
publish, or preparation of the manuscript.

**Competing interests:** The authors have declared
that no competing interests exist.

stimulus dependencies of gamma rhythm and show that this arch shaped waveform is obtained only in one of those models. Further, the waveform shape is dependent on the operating domain of the system. Therefore, shape analysis provides additional constraints on cortical models and their operating regimes.

## Introduction

Gamma rhythm refers to oscillatory neural activity in the 30–70 Hz range that changes in response to different stimuli and cognitive states [1]. In the primary visual cortex (V1), the gamma rhythm has been studied extensively using achromatic gratings, wherein gamma power and peak frequency have been shown to vary systematically with the properties of the grating [2,3]. For instance, peak frequency of gamma increases with the contrast of gratings [2,4] and, gamma power increases and frequency decreases with stimulus size [5,6]. More recently, chromatic stimuli have also been explored, which, for low-wavelength (reddish) hues, can generate huge gamma oscillations that are an order of magnitude stronger than gamma produced by achromatic gratings [7–9]. Further, these plain colored blobs of different sizes exhibited trends in gamma and firing rate analogous to that of achromatic gratings of different sizes [8].

Gamma oscillations are thought to reflect the push-pull activity of interconnected excitatory and inhibitory neurons, which has been demonstrated in different spiking network models [10–13]. However, since such large-scale network models have several parameters to be tuned and can become hard to interpret, simplified population rate models are sometimes used. Although these models have limitations because of fewer free parameters, they are analytically more tractable. One pioneering model was proposed by Wilson and Cowan (WC) [14], in which excitatory and inhibitory neurons were grouped into populations, and the dynamics of these populations were characterized. Variants of WC models have recently been used to explain some properties of the gamma rhythm. For example, based on the observation that gamma rhythm appears in short bursts and its phase does not vary linearly with time [15–17], a WC based model was proposed in which the activation function (input-output relationship) was linear but driven by Poisson noise [17,18]. A variant of this model by Jia, Xing and Kohn (JXK) [2], in which an additional global excitatory population was added, was used to explain the stimulus dependence of gamma oscillations. On the other hand, Jadi and Sejnowski (JS) [19] used a WC model with a non-linear (sigmoidal) activation function and showed that constraining the model to operate in an inhibition stabilized mode [20] and the inhibitory population to operate in superlinear domain can also reproduce the size and contrast dependence of gamma rhythm. In this study, we focus on JXK and JS models since these are both WC based models which can explain the size and contrast-based stimulus dependencies. More complex rate models that explain stimulus dependencies and potentially also the gamma shape are elaborated in the Discussion section.

Although both JXK and JS models can explain the stimulus dependence of gamma, the shape of the rhythm likely depends on the presence (and type) of non-linearity. The shape could be non-sinusoidal (see, for example, Fig 11 of [14]), which is represented in the spectral domain as peaks at harmonics of the fundamental frequency. Such distortions have been observed in various brain oscillations such as theta rhythm (for a review, see [21]). For gamma rhythm, a visual inspection of raw traces reveals some distortion [22–24], which is also corroborated by the presence of peaks at the first harmonic of the fundamental gamma peak in the power spectra [4,5,7]. However, a quantitative study of the gamma waveform, which could

potentially constrain the type and operating regime of models, has not been undertaken. This is partly because the noise in the harmonic range could offset its phase estimates if the harmonic is not prominent enough. To address this, we studied gamma oscillations produced by presenting full-screen hues that generated salient gamma oscillations with prominent harmonics [7], characterized the shape, and tested models that could replicate the shape.

## Results

We collected spikes and LFP from 96 and 81 electrodes from two monkeys, M1 and M2, while they viewed full-screen color patches of different hues. Limiting our analysis to electrodes with reliable estimates of RF centers (see "Electrode selection" section in Methods and Models) yielded 64 and 16 electrodes from the two monkeys.

### First harmonic of gamma oscillation

In Fig 1A–1D, the top row shows the power spectral density (PSD; averaged across all trials and subsequently across all electrodes) of the LFP signal in subject M1 during baseline period (-500 to 0 ms from stimulus onset; black trace) and during stimulus period (250 to 750 ms from stimulus onset; color/grey trace; the color indicates the hue that was presented and grey trace represents gratings). The corresponding change in power (dB) from baseline (Fig 1A–1D bottom row) shows a prominent peak in the gamma range, along with another prominent peak near twice the frequency. Fig 1E–1H show the same plots from subject M2. We first tested whether this second peak was indeed at twice the frequency. For each electrode and stimulus, we measured the gamma peak frequency from trial-averaged baseline-corrected PSDs as the highest peak within the 30–70 Hz band, and the peak of the second bump as the highest peak occurring after 12 Hz past the identified gamma peak frequency but before 140 Hz (the identified peaks are highlighted by black crosses in Fig 1).

Fig 2 presents the medians of the frequency ratios of second bump to gamma peak frequencies for each stimulus (hues indicated by color, gratings by grey color), computed across all electrodes in each subject, with error bars indicating the standard error of the median computed by bootstrapping. The stimuli are arranged along the horizontal axis by the average gamma power across electrodes. The histogram of the median frequency ratios across stimuli showed tight clustering around 2. Such clustering was especially observed for stimuli that produced strong gamma oscillations. To test whether the ratio for a given stimulus was significantly different from 2, the frequency ratios computed in different electrodes for the stimulus were subject to a Wilcoxon signed rank test. Stimuli for which the ratios were found to be significantly different from 2 (p < 0.01) are indicated by open circles. Estimates of peak-frequency ratios for stimuli producing less gamma power were more susceptible to noise and showed larger deviations in frequency ratios, thus contributing the most to the broad tail of the histograms. However, frequency ratios were distributed more narrowly around 2 for stimuli inducing stronger gamma. The median ± 1SE ratio value for M1 was 1.95±0.014 (red and yellow horizontal lines in the histograms in Fig 2), significantly different from 2 (p = 0.018 for M1; Wilcoxon signed rank test). For M2, the median ratio was 1.95±0.018, significantly different from 2 (p = 1.01e-5; Wilcoxon signed rank test). However, note that because we computed PSDs over a 500 ms window, the frequency resolution was 2 Hz, which introduced some error in the estimated ratio. For example, if the true gamma and harmonic peaks are at 41 and 82 Hz, the estimated gamma will either be at 40 or 42 Hz, yielding a ratio of either 82/40 = 2.05 or 82/42 = 1.95. These margins of error are shown as dotted lines in Fig 2. Most of the points lay within these error margins, especially for electrodes with higher gamma power. When we restricted the analysis to the top five hues for each monkey in terms of gamma power, the

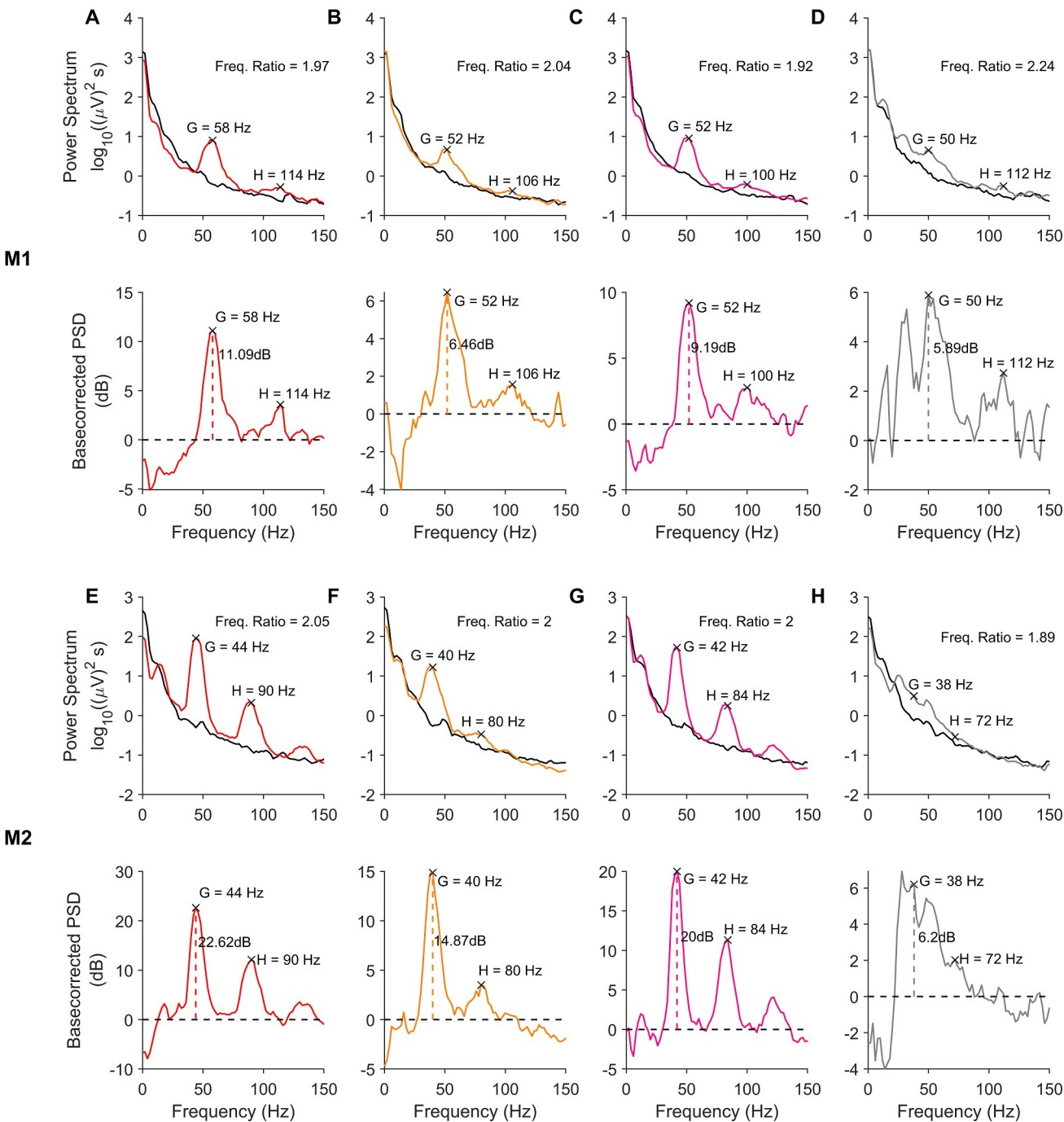

**Fig 1. Visual Stimulation induces narrowband activity in gamma and its harmonic bands.** (A-D, *top row*) PSDs for baseline (-500 to 0 ms; 0 indicates stimulus onset) and stimulus (250 to 750 ms) periods for different stimuli, averaged across trials and electrodes, from monkey 1 (M1). The baseline PSDs are plotted in black, while the stimulus period PSDs are colored (corresponding to the hue presented) or grey (for fullscreen grating). (A-D, *bottom row*) show the change in power (dB) in the stimulus period from the baseline, computed from the PSDs in the top row. The gamma power computed from the change in power spectrum is represented by vertical dashed line). The peaks in gamma range and the second bump activity are marked in each plot. (E-H) Same as A-D, for monkey M2.

ratios were 1.96±0.042 (p = 0.63; Wilcoxon signed rank test) and 1.95±0.026 (p = 0.25; Wilcoxon signed rank test). These results confirm that the second peak was indeed the first harmonic of gamma. The overall power at peak gamma and first-harmonic frequencies were also

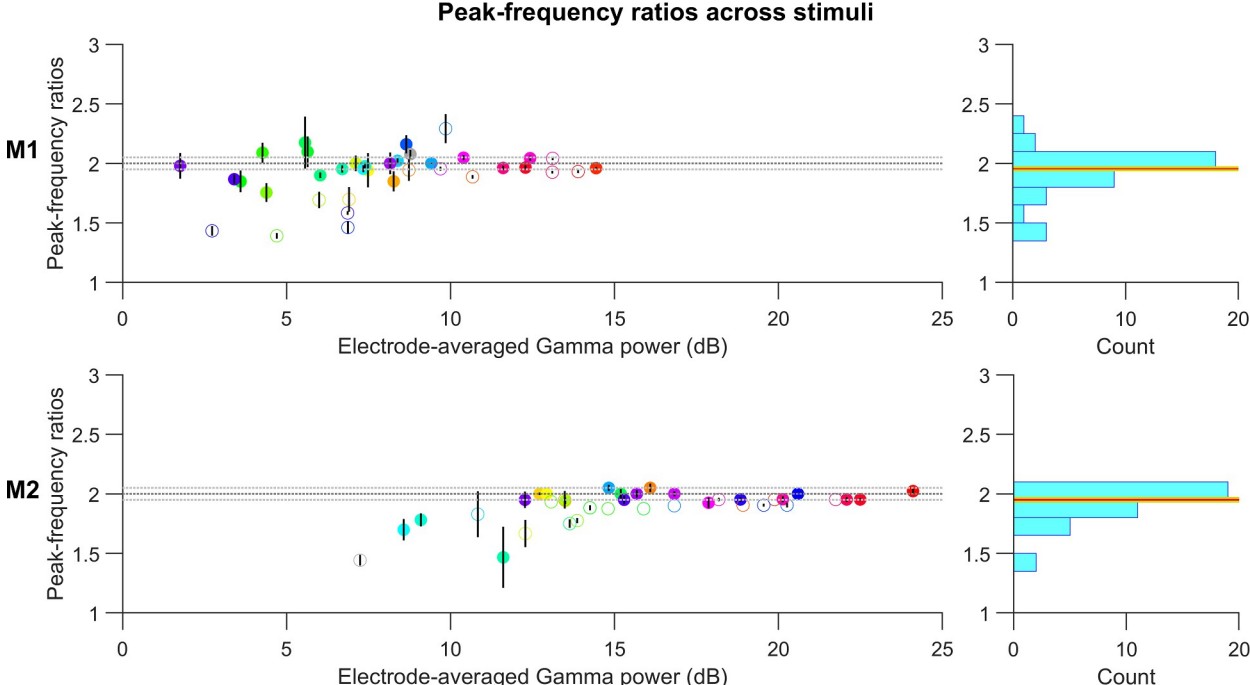

**Fig 2. Second bump in PSD corresponds to the first harmonic of gamma.** Median of peak frequency ratios of the second bump with respect to the gamma band, scattered against average gamma power for each stimulus (circles are colored to represent the presented hues; grating stimulus is represented by the grey circle). Error bars indicate the standard error of median (estimated by bootstrapping). Open circles mark stimuli whose ratios were found to be significantly different from 2 (p-value < 0.01 without any Bonferroni correction; Wilcoxon signed-rank test). The dotted horizontal lines correspond to the least-error margin 2±0.05. The histograms on the right show the distribution of the median peak frequency ratios across all stimuli. The median ± 1SE computed for this distribution are represented by the red and yellow horizontal lines respectively.

highly correlated across electrodes in each monkey (Pearson correlation coefficient was 0.85, p = 4.86e-19 for M1 and 0.86, p = 1.55e-5 for M2), and the results appeared similar if the power at first harmonic was used instead of main gamma power in Fig 2.

For hues with low gamma power, the peak frequency ratios were more frequently less than 2 in both monkeys. Inspection of their PSDs showed some overlap between the gamma and the harmonic bumps, resulting in non-zero change in power (dB) between the two. When the actual gamma and harmonic peaks were not prominent enough, the overlapping region often had higher power, leading to an overestimation of gamma peak frequency and an underestimation of the harmonic peak frequency, both resulting in a decrease in the peak-frequency ratio.

## Shape of the gamma waveform

Fig 3 shows LFP traces from example trials corresponding to each stimulus case presented in Fig 1. These traces revealed a characteristic arch shape of gamma waveform, featuring narrower troughs separated by much broader crests, hinting at a characteristic alignment of gamma fundamental and its harmonic in these cases (best observed in M2 when colored patches were shown).

To visualize how phase differences between gamma and its first harmonic components affected the shape of the summed signal, we added two sinusoids to emulate the fundamental and the first harmonic of gamma, at frequencies 45 Hz and 90 Hz respectively, and varied the initial phase of the 45Hz component (as described in Eq (2); shown in Fig 4A). Fig 4B shows

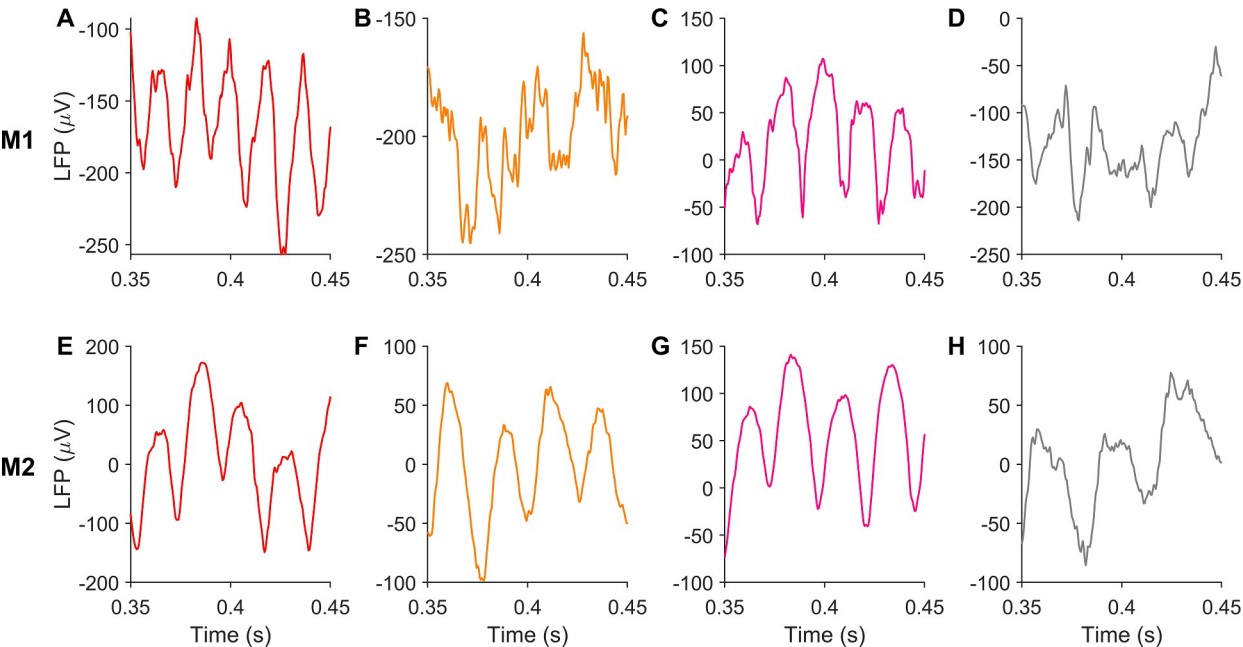

**Fig 3. Gamma waveform has consistent non-sinusoidal shape under different stimuli.** (A-D) present LFP traces for different stimuli, shown in Fig 1, from an example trial and electrode in M1. (E-H) shows LFP traces obtained from M2 for corresponding stimuli.

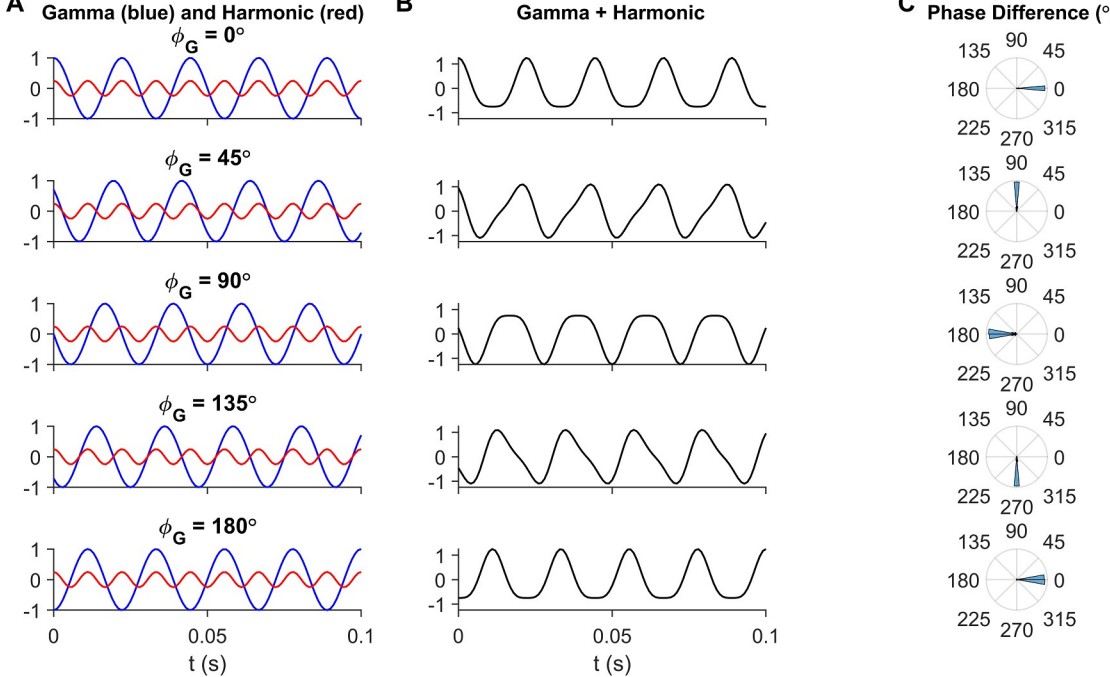

**Fig 4. Observed arch-shape of gamma corresponds to gamma-harmonic phase difference of 180°.** (A) Gamma (blue) and harmonic (red) components, mimicked using sinusoids as in Eq (2) with different initial gamma phases ($\varphi_G$) in each row. (B) Gamma waveform resulting from summation of gamma and its harmonic from the corresponding row in (B). (C) Phase difference of gamma and its first harmonic, as given in Eq (1), computed from filtering and Hilbert transforming gamma and harmonic bands from the summed waveform in (B) in each row. Note that the arch-shaped gamma observed in recordings arises from a phase difference of 180° (middle row).

the waveforms produced at each of these phases, and Fig 4C indicates the corresponding values of our phase difference measure (computed as in Eq (1)) between gamma and its harmonic. Using our convention (Eq (1)), a phase difference of 180 degrees (Fig 4A–4C row 3) gave the desired shape as troughs of both sinusoids aligned to produce a steeper overall trough.

## Gamma waveform is similar for different hues

We considered, in each electrode, the mean gamma-harmonic phase difference in each trial estimated over the stimulus period. In Fig 5A, we show the distribution of trial-averaged gamma-harmonic phase differences in all electrodes, for each stimulus (indicated by color), arranged horizontally in order of average gamma power produced. For each stimulus, the trial-averaged phase differences from all electrodes were subject to a Rayleigh test, and the stimuli with non-uniform phase difference distributions (p-value < 0.01) are indicated by filled circles. The circles are centered at the mean of the trial-averaged phase differences for all electrodes and the error bars in each case represent the 95% confidence interval of the corresponding mean phase differences. The histogram of these mean phase differences across stimuli with non-uniform phase distributions (filled circles) exhibited a peak near 180° in both subjects. Specifically, the phase differences at higher gamma generating stimuli were distributed close to 180° in both subjects, as predicted from the arch-shaped waveforms (Fig 4). Stimuli that produced low power had high dispersion of phase differences, likely originating from higher

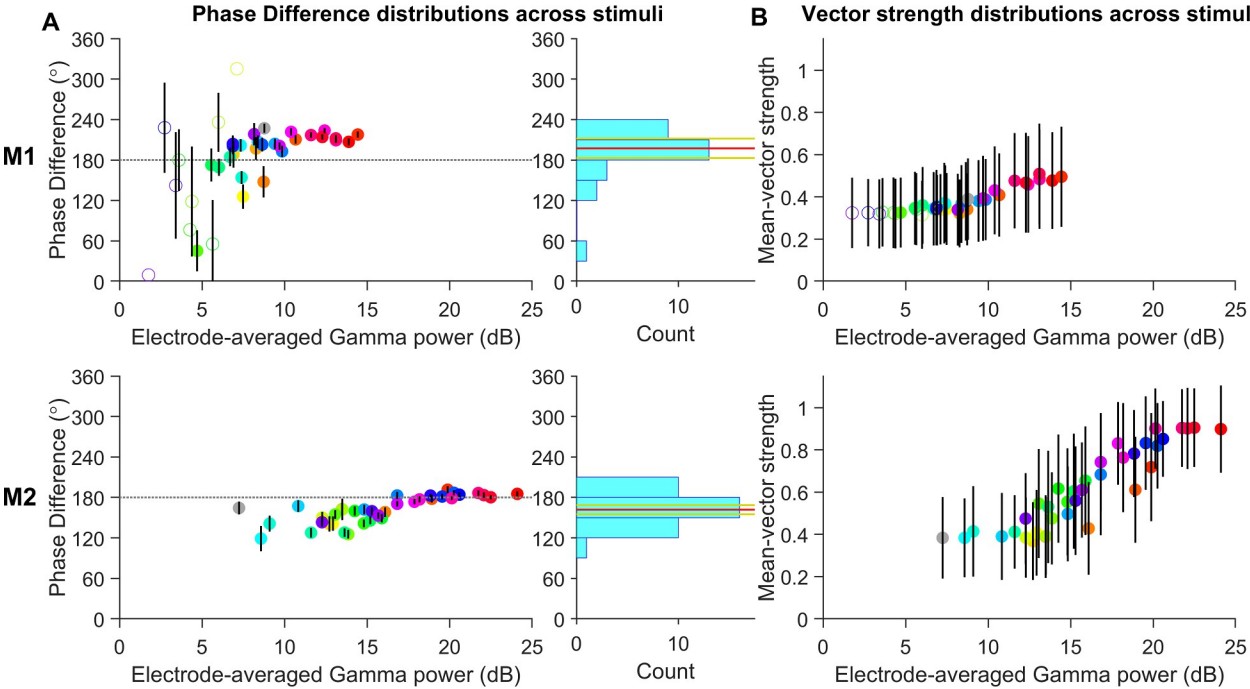

**Fig 5. Gamma-harmonic phase differences in different stimuli are concentrated near 180°.** (A) Circular mean of trial-averaged gamma-harmonic phase differences (across all electrodes) are scattered against the average gamma power for each stimulus (circles are colored to represent the presented hues; grey for grating stimulus). For each stimulus, the pool of trial-averaged phase differences from all electrodes is subject to Rayleigh test of non-uniformity to check if phase-difference is consistent across electrodes so that circular mean estimates are reliable (p-value < 0.01). If found reliable (even though it could be different from 180°), the stimulus is represented by a filled circle. Error bars represent the 95% confidence interval of circular mean. The histograms on the right show the distribution of the electrode-averaged phase differences of stimuli with reliable circular mean estimates (filled circles). The circular mean ± 95% confidence interval of this distribution are represented by the red and yellow lines horizontal respectively. (B) The mean vector strengths for the different stimuli are distributed against the average gamma power for each stimulus, similar to (A).

influence of noise, as is visible from the larger confidence intervals in M1, resulting in some hues with nearly uniform phase difference distributions (unfilled circles; some cases lack error bars as confidence intervals could not be computed due to their near-uniform distributions). For hues that generated high gamma power, gamma-harmonic phase differences were concentrated slightly higher than 180˚ for M1 and close to 180˚ for M2. The circular mean of electrode-averaged phase difference between gamma and its first harmonic across those stimuli which showed unimodal distribution (filled circles in Fig 5A; 28 stimuli in M1 and all stimuli in M2; indicated in the respective histograms in Fig 5A by the red and yellow horizontal lines) was 197.33±14.44 degrees for M1 (194.00±17.93 degrees if all stimuli were considered) and 161.79±6.79 degrees for M2. When restricted to the five hues with the highest gamma power for each monkey, the phases were 213.94±6.56 degrees for M1 and 183.92±2.26 degrees for M2. Interestingly, although the harmonic-band activity was not saliently visible in the PSD in the achromatic stimulus used here (especially for M2), the phase analysis nonetheless revealed a consistent phase difference of 227.04±6.81 degrees for M1 (Rayleigh test; p-value = 6.10e-58) and 164.38±9.57 degrees for M2 (p-value = 2.11e-29; grey filled circle in Fig 5A). The slight difference in distributions of gamma harmonic phase-differences across the two subjects could arise from unavoidable differences in the recording depths and intrinsic biological differences.

Phase differences for stimuli producing lower gamma powers appeared slightly less than those at higher powers. For a second set of five stimuli with the highest gamma power below 10 dB for M1 and below 15 dB for M2, the phases were about 20˚ less (195.18±31.09 degrees and 143.06±17.25 degrees, respectively). To test the possibility of this reduction arising from any systematic bias in the phase analyses due to noise, we performed two tests. First, we repeated the phase analyses on simulated data (see Methods and Models; *Effect of harmonic phase on gamma waveform*) by adding Gaussian noise of different variances to the summed waveform in Eq (2). Secondly, we chose trials in M2 which had the highest gamma power, and created a synthetic signal which was a weighted average of the signal segments in the stimulus and baseline periods. In both simulations, the gamma power in the simulated signal could be reduced by increasing the variance of Gaussian noise and the fraction of baseline signal, respectively. In both tests, the phase-difference estimates obtained from our analysis were not biased by higher noise levels, but had larger confidence intervals as observed in the data at lower gamma powers.

To assess the consistency of the 'arch-shape' of the gamma waveform over several cycles in each trial, a mean vector strength was computed for each trial (see *Experimental Design and Statistical Analysis* under Methods and Models). Fig 5B shows the mean of trial-averaged vector strengths across electrodes for each hue. Higher values of vector strength correspond to greater consistency of the phase-difference (and hence the waveform shape) across different gamma cycles, with 0 indicating that each gamma cycle was a different shape and 1 indicating the same shape in every cycle. Vector strengths increased with gamma power, indicating high cycle-by-cycle waveform consistency when gamma was prominent. Stimulus cases where prominent gamma was recorded (as seen for M2), the vector strengths were close to 1 indicating that arch-shape waveforms were conserved in nearly each cycle.

## Gamma waveform in a linear population rate model with stochastic inputs

To test whether the stereotypical gamma waveforms observed in the data could be modeled, we focused on two recently developed rate models by Jia, Xing and Kohn (2013; abbreviated as the JXK model) [2] and Jadi and Sejnowski (2014; JS model) [19], both of which can explain the contrast and size dependence of gamma oscillations (see *Studying non-sinusoidal waveforms in mean-field models of gamma* under Methods and Models for detailed description of

these models and their representations of stimulus size and contrast). These models, however, differ in the mechanism of gamma generation. The JXK model produces gamma through persistent stochastic perturbations provided by noisy inputs whereas the JS model produces gamma through limit cycles without the need for time-varying inputs.

The JXK model, which is a piecewise linear model, operates as a damped oscillator in its linear domain and produces sustained oscillations by virtue of time-varying input drives generated by a Poisson process causing the responses to be stochastic as well. Hence, we simulated the model over 2 seconds repeatedly for 50 iterations and analyzed the interval from 1–2 seconds ('analysis' window). Fig 6A shows the gamma peak amplitude obtained from the average PSD of the LFP proxy across iterations for each stimulus size and contrast; Fig 6B shows the corresponding mean gamma peak frequencies. These plots show the model replicating the stimulus size/contrast effects on gamma as demonstrated by Jia and colleagues [2]: gamma frequency decreased and power increased as stimulus size increased, and gamma peak frequency increased when contrast increased. Fig 6C shows the average PSD from all iterations for different stimulus sizes (r) at a fixed contrast (c) of $10^{-0.25}$, showing a distinct bump of activity in gamma range, whose peak frequency decreases but peak power increases as we go from small to larger values of stimulus size (r). Likewise, Fig 6D shows the average PSD for different contrasts and a fixed size of 4.375. The gamma peak frequency in the LFP proxy increased with contrast as demonstrated by Jia and colleagues [2]. However, the gamma bumps were broad, as expected of a noisy pseudoperiodic signal, and no prominent harmonic band activity was discernible in the PSDs. We performed phase analysis, considering the activity in the band centered around twice the gamma peak frequency (Fig 6B) as the harmonic band activity. Fig 6E shows the harmonic amplitudes and Fig 6F shows the mean gamma-harmonic phase differences of LFP proxy traces from different iterations. We searched for stimulus size and contrasts which yielded arch-shaped gamma, or equivalently a gamma-harmonic phase differences close to 180 degrees (Fig 4C, middle) consistently across iterations. Such domains of inputs are shown by black contours in Fig 6F and are identified as 'in-regime' stimuli (see Methods and Models; *Identifying the operational input regimes using gamma-harmonic phase difference*). Such regimes were few and scattered sparsely. An example LFP proxy trace for the highest contrast condition in Fig 6D, which was in-regime, is shown in Fig 6G. The top panel shows the LFP proxy trace and the bottom panel shows the gamma (blue) and harmonic (red) band signals. The LFP traces exhibited bursts of gamma oscillations with waveforms showing varied distortions (and subsequently different gamma-harmonic alignments in subsequent cycles). However, a few isolated, steep troughs could be found occasionally within some bursts, which could be responsible for the emergence of a net 180˚ mean gamma-harmonic phase difference (see Gamma oscillations in the JXK model under Discussion for further elaboration). To quantitatively assess the consistency of gamma harmonic phase difference in subsequent cycles within each trial or iteration, the mean vector strength for gamma-harmonic phase differences were computed as done on real data (S1A Fig). Unlike real LFP data for which mean vector strengths were high (Fig 5B), the JXK results gave low vector strengths even for the contoured input combinations, indicating a less frequent and more spurious occurrence of the observed 180˚ phase difference in the gamma cycles.

## Gamma waveform in JS model

Jadi and Sejnowski [19] simulated visually evoked gamma using a firing rate model with only an Excitatory and an Inhibitory population and demonstrated gamma as self-sustained oscillations in response to steady constant input drives. As a result, any waveform shape found in a cycle of the oscillations is repeated throughout. We studied whether this model could produce

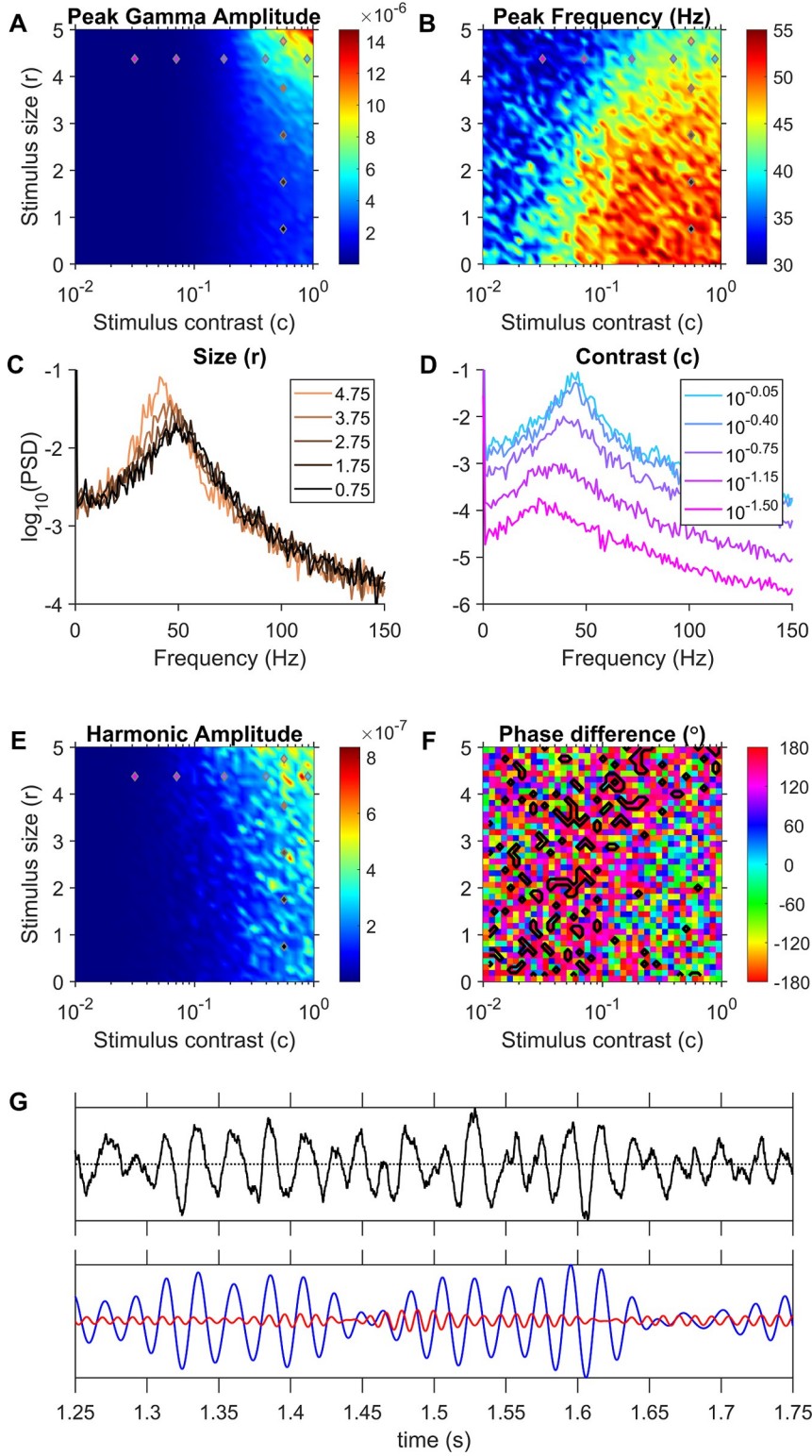

**Fig 6. Gamma properties for the JXK model.** (A) Gamma peak amplitude and (B) frequency found in the average PSD from 50 iterations at each stimulus size (r) and contrast (c) combination. (C) Average PSD of LFP proxy traces generated at fixed contrast and different sizes, across all iterations. (D) Average PSD of LFP proxy traces generated at different contrasts at a fixed size. The stimuli values that generated these PSDs are indicated in plots (A), (B) and (E) using colored markers (same colors as the plots in C and D). Input combinations used in (C) were chosen to have

constant contrast (5 vertically distributed markers) while those used in (D) had constant size (5 horizontally distributed markers). (E) Harmonic band peak amplitude in the average PSD. The harmonic is taken to be twice the gamma frequency. (F) Mean gamma-harmonic phase difference (Eq (1)) across multiple iterations, with in-regime inputs encircled by black contours. (G) Example LFP proxy activity (top panel) and its gamma and first harmonic components (bottom panel) generated at the highest contrast condition in (D) (identified as in-regime). This trace shows occasional steeper troughs (bottom trace; near 1.35 s, 1.52 s and 1.58 s timepoints) although there is no consistent trend between gamma and the harmonic.

realistic gamma waveforms by identifying the input regime (pairs of input drives to Excitatory and Inhibitory population) that would give rise to gamma-harmonic phase difference around 180° in LFP.

We implemented the model with the parameters specified in [19] for different input drives as listed in Table 1, and computed the LFP proxy in each case. Fig 7A and 7B shows the gamma peak amplitude and frequency respectively computed from the LFP proxy for each input-combination. The region within the white contour indicates an inhibition-stabilized network in which the response function of the inhibitory population operates in the superlinear regime (for details, see [19]). Fig 7C shows the PSDs of LFP proxies obtained for different inhibitory input drives and a constant excitatory drive (input-combinations marked by same-colored markers in Fig 7A and 7B), simulating the variation of stimulus size at a fixed contrast, as simulated in [19]. The PSDs demonstrate the decrease in peak frequency and increase in power of gamma in our LFP proxy signal with increasing stimulus size. To demonstrate the contrast effect, we chose a set of input-combinations where both excitatory and inhibitory input drives linearly increased with contrast (colored markers in Fig 7A and 7B). The PSDs of LFP proxy generated for increasing contrasts are shown in Fig 7D. The gamma frequency increased for higher contrasts mimicking gamma in LFP recordings [2]. Importantly, unlike JXK, this model showed power at harmonic frequencies as well.

Fig 7F plots the gamma-harmonic phase differences in the LFP proxy trace generated for each input combination. Regions in the input drives where the phase difference fell within 22.5° from 180° are enclosed by contours (Fig 7F, black contours) and identified as 'in-regime'. Interestingly, this identified regimes fell inside the region predicted by Jadi and Sejnowski [19] where inhibition was superlinear (Fig 7F, white contours), and further constrained the model to operate in more restricted regions. Note that small regions lying close to the grey region on the plot were also occasionally characterized as in-regime, but they are likely spuriously generated by simulation noise and the corresponding gamma and harmonic powers were very low (Fig 7A and 7E) and were therefore not considered. Fig 7G presents an example LFP proxy trace in the top panel, corresponding to the largest size condition in Fig 7C (in-regime), whose

**Table 1. Parameter values used in rate models.**

*JXK*

| $W_{EE}$ | $W_{EI}$ | $W_{IE}$ | $W_{II}$ | $W_{EG}$ | $W_{IG}$ | $W_{GE}$ | $\tau_E, \tau_I, \tau_G$ (ms) |
|---|---|---|---|---|---|---|---|
| 1.5 | 3.25 | 3.5 | 2.5 | 0.25 | 0.5 | 0.6 | 6,15,19 |
| MN | r | | c | | | | |
| 0 | 1 to 5 (steps of 0.125) | | 0.01 to 1 (logarithmic steps of 0.5) | | | | |

*JS*

| $W_{EE}$ | $W_{EI}$ | $W_{IE}$ | $W_{II}$ | $\tau_E, \tau_I$ (ms) | | | |
|---|---|---|---|---|---|---|---|
| 16 | 26 | 20 | 1 | 20,10 | | | |
| $m_E, m_I$ | $\theta_E, \theta_I$ | $I_E, I_I$ | | | | | |
| 1,1 | 5,20 | 0 to 20 (steps of 0.5) | | | | | |

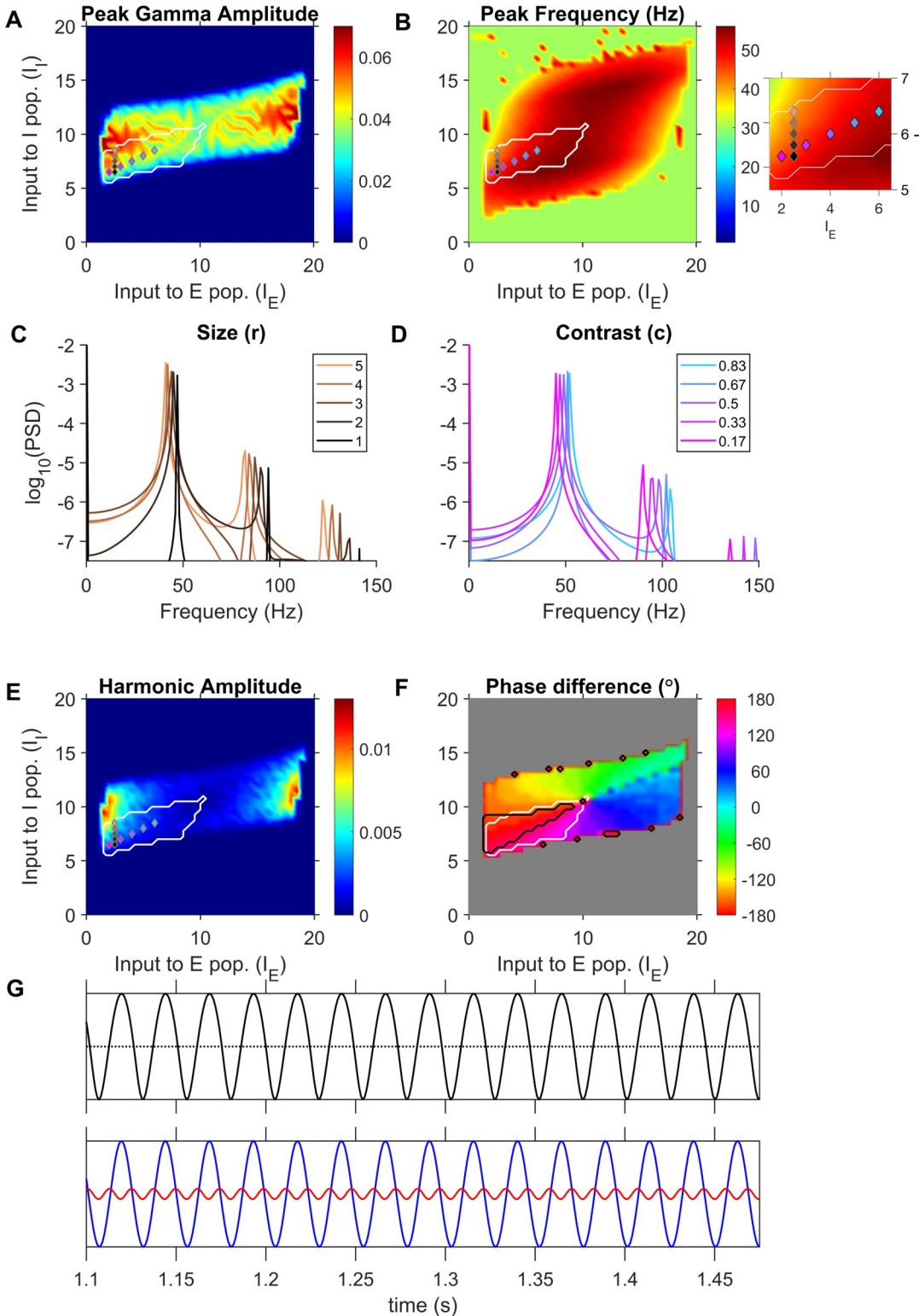

**Fig 7. JS model produces sustained arch-shaped gamma oscillations within the previously identified superlinear inhibitory regime.** (A) Peak gamma amplitude and (B) Peak gamma frequency in the LFP proxy generated by the JS model for each input-drive combination. (C) PSD of LFP proxy generated for different stimulus sizes. (D) PSD of LFP proxy generated for different contrast. Input combinations used in (C) and (D) are indicated by markers, with same colors as traces, in (A), (B) and (E). The Inset on the right of (B) shows the same plot over a more restricted range for better visualization of input combinations. Input

combinations used in (C) were chosen to have constant $I_E$ (5 vertically distributed markers; the increase in size modelled as an increase in $I_I$) while those used in (D) simultaneously varied in $I_E$ and $I_I$ (5 markers distributed obliquely; the increase in contrast modelled as a linear increase in $I_E$ and $I_I$). (E) Harmonic amplitude identified from the PSD of LFP proxy for each stimulus condition. (F) Phase difference of gamma and its harmonic (Eq (1)) computed for each simulated input drive pair. The black contours encircle input drives identified to be 'in-regime' from their phase differences. The white contour in each figure enclose the input domain identified by Jadi and Sejnowski (2014) to replicate gamma power increase and peak frequency decrease in response to increasing stimulus size, and it can be seen to enclose a significant portion of the 180˚ phase difference regime (black contour). (G) LFP proxy activity trace corresponding to the largest stimulus size shown in Fig 7C. The dotted horizontal line shown is equidistant from the minima and maxima of the traces. A sinusoidal oscillation would have the same crest and trough width on this line. The bottom panels show the gamma and its first-harmonic components, plotted in blue and red respectively, filtered from the LFP proxy traces.

gamma fundamental (blue) and first harmonic (red) components are shown in the bottom panel. The troughs of gamma and harmonic were indeed aligned in each cycle, adding up to a wider crest and a sharper trough overall, resulting in mean vector strength of 1 (S1B Fig). However, we note that the overall magnitude of the harmonic was smaller than what we observed in real data. This could be a general limitation of mean field models, as discussed later.

The LFP proxy used here involves both $r_E$ and $r_I$ with equal contributions, as suggested by a computational modelling study of LFP proxies by Mazzoni and colleagues [10] (see *Relationship between firing rates and LFP* under Discussion for further details and discussion of other proxies). However, the contributions to LFP by different populations could vary with depth of recording, proximity to the neurons since they determine the relative distribution of different synapses in the recording neighborhood [25]. To test whether the ability of the models to generate arch-shaped waveforms depends on the relative contribution of the two populations, we tested proxies containing contribution from only one population. S2 Fig shows the gamma-harmonic phase differences and regimes found in both the models for these proxies. S2A Fig show the phase differences in the JXK model using -rE as LFP proxy and S2B Fig shows the results using -rI. As before, no connected region of in-regime stimuli could be found (similar to the case in Fig 6F). S2C and S2D Fig show the phase differences found in the JS model using these LFP proxies. While the location of in-regime inputs identified was different, the superlinear regime was still found to contain a connected region of in-regime inputs in either case, although the operating regions were very small now. Nevertheless, this suggests that an LFP proxy chosen as any non-negative linear combination of -rE and -rI, would give rise to 'in-regime' inputs within the superlinear regime of the JS model.

## Discussion

We show quantitatively that gamma oscillations produced by different hues possess a distinctive arch-shaped waveform, which leads to a distinct peak at the first harmonic of gamma in the PSD, and a specific phase relationship between gamma and its harmonic. Further, we show that a linear, stochastically forced model proposed by Jia and colleagues [2] does not produce distinct harmonics with consistent phase relationship and hence does not retain a specifically-shaped gamma waveform. On the other hand, a non-linear, self-oscillating model proposed by Jadi and Sejnowski [19] produces prominent harmonics, and in a subdomain of inputs within the previously identified superlinear regime, the model generates the observed gamma waveform as well.

One important point to note is that this arch shaped gamma waveform with prominent harmonics were characterized for hue induced gamma, whereas most previous reports have used achromatic stimuli. In the achromatic stimulus used here, harmonics were not salient, especially for M2. However, although earlier experimental studies using achromatic gratings did

not describe or quantify the gamma waveform, visual inspection of raw LFP traces containing gamma bursts reveal a similar shape in some studies (Fig 1A of [26], Fig 2 of [23] and Fig 2 of [17]) showing that the arch-shape of gamma is observed for at least some achromatic stimuli. The arch shape described here is also visible when colored natural images were presented (Fig 2 of [27] and Fig 2 of [9]). Interestingly, some papers (Fig 1 of [5]; Fig 9 of [26]) have shown traces with an inverted arch shape. It is unclear why this is the case; potential reasons could be due to differences in the position of the reference wire, cortical depth of recording, or differences in the conductance level. For example, arch-shaped gamma waveform is observed in biophysical ING-PING and spiking models as well (Fig 4 of [10]; Fig 4 of [28]), in which non-sinusoidal gamma arises from the dynamics of the voltage-gated conductances, and the specific shape produced by the model varies when the conductances are varied (effects of conductances on gamma shape could be operating through similar shifts in input/response regimes as shown here). LFP waveform also changes with the recording depth (Fig 4B versus 4C of [10]). Non-sinusoidal waveforms are common in neural rhythms, such as the sawtooth-shaped theta rhythm in the hippocampus and comb-shaped beta rhythm, arising from a variety of different biophysical phenomena [21].

Generalizing the arch shape and the associated phase-difference between gamma and its first harmonic observed in the fullscreen iso-luminant chromatic stimuli used in this study to other visual stimuli would be challenging, especially for stimuli that produce less gamma powers such as achromatic gratings of smaller sizes. Indeed, for stimuli such as small achromatic gratings that do not produce strong gamma but produce high firing rate (leading to high broadband 'pedestal'), estimation of the phase of the gamma or harmonic may not be possible at all, even though the two models used in this study had been designed to explain the firing rates and gamma observed when achromatic gratings were presented. This is because unlike quantities such as change in power, for which the 'pedestal' can be subtracted out by either using the power spectral density at baseline period (as done here) or by explicitly modeling the pedestal [29], the overall phase at a particular frequency has contributions from both the true gamma signal and the broadband pedestal. Phase analysis is therefore susceptible to errors and increased variability caused by high broadband noise, especially at low gamma powers (as evident from the larger error bars at low powers in Fig 5A).

Although our results may not be generalizable to stimuli that do not produce very strong gamma, they may nevertheless be useful when comparing two different schemes or principles that produce oscillations. As discussed below, the two models use two very different principles– the JXK model is a damped oscillator which is driven by stochastic noise and therefore cannot produce gamma that is structured over long timescales, while oscillations in the JS arise from a resonance phenomenon which can maintain a consistent shape over multiple cycles. Therefore, by showing that gamma oscillations have a consistent arch shape (even for some classes of stimuli), a resonant type architecture can be favored over a noise-driven architecture.

## Gamma oscillations in the JXK model

The JXK model, like another model proposed by Kang and colleagues [18], explains the "bursty" nature of gamma oscillations, which was shown in previous studies [15,17]. These studies observed that gamma band power varied during a trial, with short intervals (~130 ms) of high gamma power, interspersed by durations of no prominent power. Based on these, Xing and colleagues [17] suggested that gamma rhythms were the outcome of cortical network resonating to the stochastic component in the inputs. During periods in a trial when stochastic variations in input are less, the cortical network would tend to asymptotically stabilize and oscillations would get weaker over time. Xing and colleagues [17] demonstrated this

mechanism in a piecewise linear asymptotically stable model with noisy inputs. The JXK model [2] emulated gamma burst-activity exhibiting the observed dependencies of gamma power and frequency on stimulus parameters such as size and contrast by varying input drives and recurrent excitation to the neural populations. However, the JXK model does not produce a consistent waveform shape, because despite the presence of a rectification component, the model operates mostly in a linear (post-rectification) domain for all inputs simulated. In this domain the step response would be sinusoidal oscillations damping exponentially over time. Although the presence of Poisson inputs distorted the waveform and prolonged oscillations for a longer duration, these distortions are not consistent across different cycles of oscillation. This was reflected as wide band of activity around the gamma band in the PSD (Fig 6C and 6D), and a lack of discernible harmonic activity. Therefore, while these models adequately explained the bursty nature of gamma oscillations, it is not surprising that they failed to produce a specific gamma waveform.

However, a recent study has shown that the duration of gamma 'burst' activity reported in Xing and colleagues [17] could have been underestimated [30]. This error arises because the spectral estimator that is used to compute gamma power is itself noisy, causing the rhythm to appear bursty in the spectral domain (see [30] for details). Chandran KS and colleagues [30] used Matching Pursuit algorithm with a Gabor dictionary, which chooses the best-matching template to estimate the duration of a gamma burst in the time domain itself to show that gamma bursts during presentation of achromatic grating were longer than reported earlier (median of ~300 ms, although the mode was ~100 ms). Along with the results presented here, this suggests that the gamma rhythm could have more temporal regularity than what is predicted by these stochastic models.

However, we note that the non-sinusoidal nature of gamma rhythm described here could potentially be introduced in the JXK model by changing the activation function. Specifically, if a non-linear activation function is used in the place of the piece-wise linear rectifier, the step response of the system would itself be a specific non-sinusoidal waveform in subsequent cycles of the resulting damped oscillation. When stochastic perturbations are added, the resultant bursts of oscillatory activity could contain both irregular distortions caused by the noisily fluctuating inputs and more regular distortions induced by non-linearity at specific parts of the trajectory in each cycle. Therefore, although we show that our results are reproduced by the JS model within the superlinear regime, it is possible that the JXK model with a suitable activation function can also replicate our findings. Further, in the JXK model, we have used Poisson noise as in the original paper [2] to study the inherent propensity to produce arch-shaped waveforms. However, a more realistic input would be colored noise [31], and it would be interesting to study the modulatory effects on the waveform shape in different cycles. Furthermore, the JXK model, despite having a rectification in its activation, operates in the linear region in almost all gamma cycles. A substitution of a smoother non-linearity with similar slope could still induce distorted waveforms while still requiring noisy inputs to sustain oscillations. Indeed, while the JS model generates the proper gamma shape within a sub-regime, the duration of gamma is as long as the duration of the stimulus itself. Therefore, some stochasticity may be necessary to replicate all features of gamma rhythmicity, including duration and shape. We therefore tried several different non-linearities such as power law rectification or sigmoid in the JXK model (see S3 Fig for results on power law non-linearity) but found that the model still retained its forced oscillator nature. The location and distribution of the sparse 'in-regime' input combinations varied, but the waveform did not show consistent arch-shaped gamma, suggesting spurious origins of this phase-difference (as revealed by low mean vector strengths in S3 Fig). A more thorough analysis is required to determine the extent to which consistent 'arch-shaped' waveforms can emerge in non-linear forced oscillatory models.

In the JXK model, there were a few inputs that satisfied our phase-difference criteria for 'in-regime' behavior. However, a close examination of the 'in-regime' trace (example in Fig 6G) revealed that, in addition to the irregularly distorted gamma cycles, these inputs had occasional isolated steep troughs. These steep troughs could be approximated by a negative impulse, which could be decomposed as a series of sinusoids with their troughs aligned, resulting in sporadic 180-degree phase-difference estimates. However, these sharp troughs and the ensuing spurious phase differences were rare within a trial and hence resulted in predominantly low trial-averaged direction vector strength as shown in S1A Fig. Since these distortions are of lower frequency than gamma, the JXK model does not operate 'in-regime' for any input combination.

## Gamma oscillations in the JS model

The JS model is a two-dimensional WC model that operates in ISN mode, where the strong recurrent excitation would lead to instability if the inhibitory population were inactivated [20]. Using a WC model operating as ISN, Tsodyks and colleagues [20] were able to model firing rate modulation and phase shift of CA1 interneurons in theta rhythm cycles, and further demonstrated a paradoxical behaviour where reduction in external drive to inhibitory population resulted in a transient increase in inhibitory activity. In V1, Ozeki and colleagues [32] modelled surround suppression as an increase in inhibitory drive and found that the ISN model showed a simultaneous reduction in network excitation and inhibition, as observed in their experiments.

Jadi and Sejnowski [19] modelled gamma rhythm as limit cycles generated for a range of inputs by a local instability and observed that the observed stimulus dependencies of gamma amplitude and frequency could be simulated while the model operated in a certain range of inputs, which resulted in the local activation of inhibitory population to be strongly superlinear. Superlinear activation in individual neurons have been used to explain normalization and size-tuning in the Supralinear stabilized network (SSN) model [33]. Note that while the SSN model specifies power law activations for both the excitatory and inhibitory cells, the 'Superlinear' regime of JS requires only that, about the steady-state activity, excitatory population activation either be weakly superlinear or sublinear when the inhibitory population is strongly superlinear. Stronger super linearity of inhibitory population could be attributed to their higher firing rates and a higher convergence of lateral excitatory inputs onto inhibitory cells.

In the current study, we could observe that the superlinear regime contains a subregion where the waveform shape of limit cycles is the same arch shape observed during presentations of our fullscreen hues and gratings. Further, we note that for proxies $-r_E-r_I$ and $-r_E$, the in-regime input combinations include the higher values of inhibitory drive ($I_I$) (see Figs 7 and S2), which in the model, corresponds to surround suppression during presentation of larger size stimuli such as the full-field stimuli used in our experiments.

## Limitations and future directions

While the LFP simulated in the JS model exhibited the desired gamma-harmonic phase differences and size-contrast effects, there are several limitations. First, the regime for the arch shape was obtained for a highly restrictive set of inputs. Second, the JS oscillations showed very weak distortions from a sinusoid (low harmonic power). Finally, the results were tested only for a particular type of activation function as described in the JS model. We found that changing the type of activation function had a drastic effect on both JS and JXK models. While it is possible that another type of activation function can produce stronger harmonics over a larger input

space, we have tried to remain as close as possible to the original models and tested whether these models can satisfy the new constraints that are imposed by the shape of gamma and the presence of harmonics.

It is likely that more complex models such as PING models [10] and rate models with synaptic dynamics [11] can explain these results, since they often exhibit strong arch-shaped waveforms (see Fig 4 of [10] and Figs 1 and 2 in [11]; the rate and synaptic variables show inverted arch-shape with sharp crests, which yield arch-shaped gamma in our LFP proxy from Eq (1)). Indeed, a typical weak PING model involves the activation of the E population that then excites the I population within a few milliseconds, and the resulting inhibition shuts down both E and I cells, with the cycle repeating once the E cells recover from the inhibition and fire again [12,34]. Some experimental studies have indeed shown that spikes tend to occur just before the trough of gamma [35–37], with the pyramidal cells leading the interneurons by a few milliseconds [38–40]. In the JS model as well, we found that the E population led the I population by about 50˚ or 2–3 ms in the superlinear regime. Further, the results were qualitatively similar when just E or I or the sum of both were used as the LFP proxy (Figs 7 and S2).

## Relationship between firing rates and LFP

JXK and JS models used excitatory cell activity to study the power and frequency trends of gamma. Here, we converted these firing rates to a "proxy" LFP to be compatible with the real data. Because the extracellular potential is generated due to the spatial separation of transmembrane currents (generating dipoles or multi-poles), as well as alignment of such dipoles across neurons, LFPs are thought to mainly reflect the transmembrane currents of pyramidal neurons [41]. Hasenstaub and colleagues [38], in their in-vivo study of ferret prefrontal cortex, found that the postsynaptic potentials and the firing of regular-spiking excitatory neurons and fast-spiking interneurons were both synchronized to the extracellular gamma rhythm, but the activation of interneurons was more aligned with the troughs of the oscillation. A modelling study [10] simulated a Leaky Integrate-and-Fire (LIF) network and computed LFP by injecting the simulated synaptic currents on corresponding synapses distributed over a population of morphologically realistic neuron models. The LFP, thus simulated, was compared against various outputs of the LIF network (such as firing rates, membrane potential and synaptic currents). They found that the sum of absolute values of synaptic currents (with both AMPA and GABA currents weighted nearly equally) served as the best proxy for LFP. This paradox of both depolarizing (sink) and hyperpolarizing (source) postsynaptic currents adding up in the same polarity could be explained considering the spatial organization of the synapses. In a cortical pyramidal cell, inhibitory synapses are typically concentrated towards the soma, whereas excitatory synapses are distributed less densely along the apical and the basal dendrites. In such an arrangement, a large proportion of the depolarizing and the hyperpolarizing postsynaptic currents would themselves constitute 'synaptic' dipoles [25], whose effect on the LFP would be in the polarity of the source or sink closest to the electrode. Consequently, the polarity (sign) of LFP proxy would vary with recording depth (see Fig 4 in [10]). The arrays in our dataset were 1 mm long and therefore are expected to be in superficial/granular layers where the firing of both E and I cells are expected to produce a downward deflection in the LFP [10]. Hence in our simulations, the LFP proxy was taken as the sum of excitatory and inhibitory contributions, both with a negative polarity.

## Gamma and Harmonics vs slow/fast gamma

Murty and colleagues [3] showed that fullscreen achromatic gratings produced two different oscillations simultaneously–a fast gamma (40–70 Hz), which has been observed previously for

smaller sized gratings, and a slow gamma (20–40 Hz) which is prominent only for large stimuli. Even though this also generates two bumps in the PSD, fast gamma is not a harmonic of the slow gamma. First, the peak frequency of fast gamma was not twice that of slow gamma. Second, slow and fast gamma were not co-tuned: they had distinct orientation, contrast, temporal frequency and size tuning preferences.

The arch-shape shown here poses additional problems for phase coding schemes in which spike position relative to the gamma phase is used to code information [36,42], because spikes tend to occur near the trough of the rhythm and having a sharper trough reduces the operating range. Other studies have proposed that gamma oscillations may not play a role but could be a useful marker/indicator of cortical processing [24]. In that framework, properties such as shape and duration along with stimulus tuning could provide additional clues about the underlying circuitry.

## Methods and models

### Ethics statement

All monkey experiments were carried out in adherence to guidelines approved by the Institutional Animal Ethics Committee of the Indian Institute of Science-Bangalore and the Committee for the Purpose of Control and Supervision of Experiments on Animals.

### Data acquisition

We used LFP data recorded by Shirhatti and Ray [7] from V1 of two female macaques, whom we refer to as M1 and M2 (correspond to M1 and M3 in the earlier paper). Data from the third monkey were not considered in this paper since it showed relatively weaker gamma and harmonic power and very strong broadband high-gamma power, which caused systematic biases in the harmonic peak-frequency estimation. However, even for this monkey, the phase relationship between gamma and its harmonic (as shown in Fig 2B) was around 180˚, similar to the other two monkeys.

Both monkeys were fitted with a titanium headpost and trained to perform a visual passive fixation task, after which a Utah array (96 and 81 electrodes for M1 and M2) was implanted in V1 (details of the surgery and implants are provided in [7]). The raw signals from microelectrodes were recorded using the 128-channel Cerebus neural signal processor (Blackrock Microsystems). LFP was obtained by filtering the raw signals online between 0.3 Hz and 500 Hz (Butterworth filters; first-order analog and fourth-order digital respectively), and recorded at 2 kHz sampling rate and 16-bit resolution. No further offline filtering was performed on this data before analysis.

Multiunit activity was also extracted from the raw signal by filtering online between 250 Hz and 7.5 kHz (Butterworth filters; fourth-order digital and third-order analog respectively), and subjecting the resultant signal to an amplitude threshold of ~5 SD of the signal. The recorded units were found to have receptive fields located in the lower left quadrant of the visual field with respect to fixation in both monkeys, at an eccentricity of $\sim 3˚–4.5˚$ in M1 and $\sim 3.5˚–4.5˚$ in M2. Full-field iso-luminant hues did not drive the neurons well, and therefore we did not get usable spiking activity from most electrodes (see S1 Fig of [7] for description of spiking activity). In our current study, all the analyses were performed only on LFP data.

### Experimental setup and behavior

During the experiment, the monkey was seated in a monkey chair with its head held stationary by the headpost. The monkey viewed a monitor (BenQ XL2411, LCD, 1,280 × 720 resolution,

100 Hz refresh rate) placed ∼50 cm from its eyes. The monkey and the display setup were housed in a Faraday enclosure with a dedicated grounding separate from the main supply ground to provide isolation from external electrical noise. The monitor was calibrated and gamma-corrected using i1Display Pro (x-rite PANTONE) to obtain a mean luminance of 60 cd/m$^2$ on its surface and to obtain a gamma of unity for each of the three primaries of the color gamut, which had the following CIE chromaticity xy coordinates: red, (0.644, 0.331); green, (0.327, 0.607); blue, (0.160, 0.062). The white point was at (0.345, 0.358).

Each monkey performed a passive fixation task, which required them to fixate at a small dot of 0.05˚–0.10˚ radius at the center of the screen throughout the trial (3.3 or 4.8 s duration; fixation spot was displayed throughout). Each trial began with fixation, following which an initial blank grey screen of 1,000 ms was displayed, and then, two to three stimuli were shown for 800 ms each with an interstimulus interval of 700 ms. The monkey was rewarded with juice for maintaining fixation within 2˚ from the fixation point. Trials in which fixation was broken were not considered in our analyses. Eye position data was recorded as horizontal and vertical coordinates using the ETL-200 Primate Eye Tracking System (ISCAN) and monitored throughout the task using custom software running on macOS, which also controlled the task flow, generated stimuli, and randomized stimuli presentation. Although the monkeys were required to maintain fixation within 2˚ of the fixation spot, they were able to hold their fixation well within these limits (standard deviation < 0.4˚ across all sessions for all monkeys) during the task. Since the stimuli were all full screen, the small deviations are unlikely to affect the results.

## Stimuli

The stimuli consisted of 36 hues and 1 achromatic grating. The hues were equally spaced along the circular hue space of the standard HSV nomenclature (0˚ hue to 350˚ hue, where 0˚, 120˚, and 240˚ represent red, green, and blue respectively), which were displayed full screen and at full saturation and value. The achromatic grating was at an orientation of 90˚ and had a spatial frequency of 4 cpd for M1 and 2 cpd for M2. These grating parameters were optimized to capture strong fast gamma with minimal slow gamma (see [3]). The full-screen stimuli, in our setup, subtended a visual angle of ∼56˚ in the horizontal direction and ∼33˚ in the vertical direction.

## Electrode selection

As with our previous report [7], electrodes were considered for analysis only if they gave consistent stimulus-induced changes and reliable receptive field estimates across sessions, determined by a receptive field mapping protocol that was run across multiple days [43]. Further, we discarded signals from electrodes with unusable or inconsistent signals, a high degree of crosstalk with other electrodes, or impedances outside the range of 250–2,500 kΩ for monkey M1 and 125–2,500 kΩ for M2. This resulted in 64 and 16 usable electrodes for M1 and M2, respectively.

## Data analysis

Stimulus presentations with excessive recording artifacts (<5.9% and <5.0% of presentations of each stimulus in M1 and M2) were discarded for each session. The typical artifacts were large deflections in the recorded traces observed across electrodes, usually accompanying occasional physical movements from the subjects (jaw movements, for instance). A trial was rejected from our analyses if the corresponding LFP trace showed deflections about the mean greater than 6 times the standard deviation of the traces during the stimulus period. This

process removed prominent artifacts and yielded 19.9±6.2 repeats in M1 and 20.2±1.0 repeats in M2 per stimulus.

## Spectral analysis of LFP

For each stimulus, LFP recorded from -500 to 0 ms from stimulus onset was taken as the 'baseline period' and 250 to 750 ms from stimulus onset was taken as the 'stimulus period' to avoid the transient responses to the stimulus onset. This yielded a frequency resolution of 2 Hz in the Power Spectral Density (PSD). PSD was computed using the Multitaper method using the Chronux toolbox [44], with three tapers. The change in power was calculated as 10 times the difference between base-10 logarithm of PSDs at stimulus period and baseline period, expressed in decibels (dB). Estimation of peak frequencies was done on these baseline-corrected PSDs.

Gamma range was taken as 30–70 Hz, and 'gamma peak frequency' was estimated as the highest peak within this range. In most stimulus conditions tested, a discernible 'second bump' was observed in the baseline-corrected PSD. The peak frequency of the second bump was estimated as the highest peak occurring beyond 12 Hz after the estimated gamma peak frequency up to 140 Hz, to exclude higher frequency bumps.

## Analysis of gamma and harmonic phases

To compute the phase difference of gamma and its first harmonic, gamma and harmonic signals were extracted from LFP during the stimulus period by bandpass filtering using separate Butterworth filters (zero-phase; order 4). The passband for gamma was 20 Hz wide, centered around the gamma peak frequency, identified from trial-averaged PSD for a given stimulus and an electrode. The passband for the first harmonic of gamma was also 20 Hz wide but centered around twice the corresponding gamma peak frequency. The phases of these signals ($\varphi_{gamma}$ and $\varphi_{harmonic}$) were then computed using Hilbert transform. The phase difference between gamma and its harmonic was calculated as:

$$\text{Phase difference} = |2^*\varphi_{gamma} - \varphi_{harmonic}| \tag{1}$$

The above phase difference estimate was computed at each timepoint within the stimulus period of every trial for each electrode. Note that the 20 Hz passband for gamma and its harmonic-band filters is wide enough so that errors in gamma peak-frequency detection do not affect the phase difference estimates.

## Effect of harmonic phase on gamma waveform

To illustrate the effect of the gamma-harmonic phase relationship on the shape of gamma waveform, gamma and its first harmonic were mimicked using sinusoids and the initial phase of gamma ($\varphi_G$) was varied. The formulation of gamma and harmonics were as follows:

$$\text{Gamma wave, } G = \cos(2\pi\,45\,t + \varphi_G)$$

$$\text{Harmonic wave, } H = 1/4^*\cos(2\pi\,90\,t)$$

$$\text{Sum of gamma and harmonic} = G + H \tag{2}$$

These waveforms are displayed in Fig 4A and 4B along with the phase difference (Fig 4C) obtained by applying Eq (1).

## Experimental design and statistical analysis

First, we tested our hypothesis that the second bump in PSD accompanying gamma was indeed its harmonic by computing the ratio of harmonic to gamma peak-frequencies, obtained from the trial-averaged change in power (dB) from baseline spectra for each electrode. The ratios were subsequently subjected to a non-parametric Wilcoxon signed rank test (Null hypothesis: median ratio = 2). The standard error (SE) of the median of the gamma-harmonic frequency ratio was estimated by bootstrapping over N iterations (where N is the number of datapoints). This involved random sampling with replacement of the ratio data N times and estimating their median each time, which resulted in N medians, whose standard deviation (SD) is reported as the standard error (SE).

Circular statistics on the gamma-harmonic phase difference data were computed in Matlab using the Circular Statistics toolbox [45]. The mean phase differences are reported as MEAN ± CI in Fig 5A, where MEAN is the circular mean and CI is the 95% confidence interval of MEAN, under a Von Mises distribution (implemented in *circ_mean* and *circ_confmean* functions of the Circular Statistics toolbox).

To validate that the distribution of gamma-harmonic phase differences was non-uniform and, thus, assess the validity of circular mean estimates in Fig 5, we subjected trial-averaged phase-differences gathered from all electrodes for a given stimulus to a Rayleigh test of non-uniformity (Null hypothesis: uniform distribution of phases). When the gamma activity recorded for a given stimulus has a specific waveform across electrodes, the corresponding distribution of gamma-harmonic phase-differences will be unimodal. We considered this to be the case when the p-value of the Rayleigh test was less than 0.01. Furthermore, we computed the mean vector strength of the gamma-harmonic phase-difference to quantify the consistency of the non-sinusoidal distortion in subsequent gamma cycles. At each time point within the analysis window in a trial, unit vectors were taken with orientation equal to the gamma-harmonic phase-difference at each time point within the analysis window and the magnitude of the average of these unit vectors was considered as the mean vector strength for the corresponding trial. The mean vector strengths across trials of a specific stimulus presentation were then averaged for each electrode. The mean vector strength corresponding to each stimulus is presented in Fig 5B as MEAN ± SE, where MEAN and SE are the average and standard deviation of the trial-averaged vector strengths across electrodes for each stimulus.

## Studying non-sinusoidal waveforms in mean-field models of gamma

To understand the factors resulting in the characteristic waveform of gamma, we investigated the gamma-harmonic phase relationship emerging in two mean-field models introduced in earlier works, which produced gamma oscillations with power and frequency trends as observed in experiments. We explored the waveform in these models by identifying input regimes that produced a first-harmonic with a specific phase difference from the fundamental of gamma, as required for its arch-shape. We assessed the presence of such regimes in two models. The first model operated linearly and produced oscillations by virtue of stochastic time-varying inputs (Jia-Xing-Kohn or JXK model; [2]), while the second model had non-linear dynamics, which gave rise to gamma frequency limit cycles in response to constant inputs (Jadi and Sejnowski or JS model; [19]).

## Jia-Xing-Kohn (JXK) model

Jia and colleagues [2] defined a linear EI rate-model and extended it by adding a Global Excitatory Population (G) to approximate recurrent excitatory feedback within V1 [2]. The model, when subjected to step input, acts as a damped oscillator but produces oscillations by virtue of

constant perturbation from the Poisson inputs. The original model, formulated in the paper, incorporates detailed stimulus descriptions, namely Masked Noise (MN) level and stimulus size (r) and contrast (c). The stimulus size parameter (r) scales the extent of Network recurrence (G) in steady-state proportionally as larger size stimuli excite a larger area of cortical cells, increasing the global feedback. The stimulus contrast (c) proportionally increases the drives to the excitatory and inhibitory populations, $I_E$ and $I_I$ respectively. In our simulations, the Masking Noise level (MN) has been set to 0, resulting in the following reduced formulation. The resultant reduced JXK model formulation is as follows:

$$\tau_E \frac{dr_E}{dt} = -r_E + W_{EE}\lfloor r_E \rfloor - W_{EI}\lfloor r_I \rfloor + W_{EG}\lfloor r_G \rfloor + E_{inp}$$

$$\tau_I \frac{dr_I}{dt} = -r_I + W_{IE}\lfloor r_E \rfloor - W_{II}\lfloor r_I \rfloor + W_{IG}\lfloor r_G \rfloor + I_{inp}$$

$$\tau_G \frac{dr_G}{dt} = -r_G + r^2.W_{GE}\lfloor r_E \rfloor$$

where $\lfloor x \rfloor = x$ *if* $x>0$, *otherwise* $\lfloor x \rfloor = 0$, and $E_{inp}$ and $I_{inp}$ approximate a Poisson process with average rates $I_E$ and $I_I$ respectively, which vary with stimulus contrast (c) as:

$$I_E = 40.\frac{c^2}{c^2 + 0.3^2} ; I_I = 32.\frac{c^2}{c^2 + 0.3^2} \tag{3}$$

Parameter values of this model are given in Table 1.

## Jadi-Sejnowski (JS) model

Jadi and Sejnowski [19] used a simple rate model to qualitatively describe stimulus size and contrast dependent modulation of gamma. The model consisted of an excitatory and an inhibitory population with sigmoidal activation, operating as an Inhibition Stabilized Network (ISN), and had constraints on the input drives to the populations to reproduce the increase in power and decrease in peak frequency of gamma with increasing stimulus size as earlier studies have observed in V1 [5,6].

The model defines the population firing rates of Excitatory and Inhibitory populations as follows:

$$\tau_E \frac{dr_E}{dt} = -r_E + \sigma_E(W_{EE}r_E - W_{EI}r_I + I_E)$$

$$\tau_I \frac{dr_I}{dt} = -r_I + \sigma_I(W_{IE}r_E - W_{II}r_I + I_I)$$

$$\sigma_P(x) = \frac{1}{1 + exp(m_P.(\theta_P - x))} - \frac{1}{1 + exp(m_P.\theta_P)} \tag{4}$$

In this model, larger stimuli were simulated by an increased inhibitory drive ($I_I$) to the population, owing to suppression from the surrounding populations [32], whereas stimuli of different contrasts could be simulated by applying co-varying drives to inhibitory ($I_I$) and excitatory ($I_E$) populations [46].

Oscillations in this model emerged after the system underwent a supercritical Hopf bifurcation at sufficiently high input drives. Since the model operated close to the Hopf bifurcation,

the amplitude and frequency of oscillations in firing rates could be closely approximated by linearization of the model (For the interested readers, we refer to Appendix-B in [19] for their detailed discussion). The authors deduced and demonstrated that the model gave rise to the observed trends in the gamma when the inputs were such that the inhibitory population was strongly 'superlinear'. This means that the summed inputs to the inhibitory population (from recurrent and external sources; argument of $\sigma_I$ in Eq (4)) must lie in a certain range of values where the activation function $\sigma_I$ curves upwards (increasing in slope with increasing summed input). For the sigmoidal activation function used in Eq (4), the summed inputs must operate in the lower half of the sigmoid. Superlinear activation of the excitatory population, on the other hand, was antagonistic and, hence, excitatory activation should not be strongly superlinear. The set of such inputs constitute the operating regime of the model, which we refer to as the 'superlinear' regime. In Fig 7C and 7D, we have presented examples of the size and contrast effects on gamma rhythm in the model. The example input combinations (indicated by markers in Fig 7A and 7B) were chosen such that they lay within the 'superlinear' regime and were varied based on the above criteria to represent different sizes and contrasts.

### Identifying the operational input regimes using gamma-harmonic phase difference

Because the gamma-harmonic relationship was studied using LFP recordings in real data while the model simulations yielded firing rates of excitatory and inhibitory populations, these firing rates had to be converted to a 'proxy' LFP. The relationship between the two is complex [10,38,41]. For simplicity, we used the negative sum of population firing rates of the excitatory and inhibitory populations in the models as a proxy for LFP, since the LFP, being the extracellular potential, could be approximated to vary inversely with the depolarization (excitability) of these populations. Furthermore, the spatial organization of the synapses are likely to produce strong dipole contributions with negative polarity to the LFP recorded in our setup. We elaborate this further in the Discussion section. We also approximated the LFP as the negative of just the excitatory or inhibitory populations, which yielded qualitatively similar results.

The environment used for simulation was Matlab 2019b (Mathworks, RRID:SCR_001622), where the models were simulated by a forward Euler method, with parameters as in Table 1. Simulations were run for a duration of 2 seconds in time-steps of 0.1 ms, on all pairs of 41 excitatory and 41 inhibitory input values sampled from the ranges specified in Table 1. In each case, PSD of the LFP proxy was computed between 1–2 seconds to avoid the initial slow transient. The gamma and harmonic peak frequencies were identified as frequencies containing maximum power in the 30–70 Hz range and twice the gamma peak frequency, respectively. For further analyses in the JS model, only those input combinations were considered for which gamma and harmonic frequency amplitudes were greater than 1e-3 and 1e-6 units respectively, to ensure phase analysis of oscillatory activity was not dominated by simulation errors or noise (grey region in Fig 7F). The LFP proxy signal was filtered using 20 Hz passbands centered at each frequency identified above. The gamma-harmonic phase difference (Eq (1)) was computed, just as for LFP in macaque data. An input point (pair of values supplied as excitatory and inhibitory input drives; stimulus parameters in the case of JXK) was said to be 'in-regime' if the gamma-harmonic phase difference was within 22.5 degrees from 180-degrees (ideal for arch shape; Fig 4). Since JXK takes stochastic inputs, to assess if there is indeed a unique gamma-harmonic phase difference at each of the mean input-drive combinations tested and whether the model can indeed exhibit in-regime behavior, we ran the simulation 50 times (analogous to 50 different trials) for each stimulus condition. In each iteration, we considered the mean phase-difference over the 1–2 second interval. For every selected input

combination, the mean phase difference over this interval in each iteration was taken and the resultant pool of 50 mean phase differences was subjected to Rayleigh test of uniformity to identify those inputs that have a 'consistent' gamma-harmonic phase relationship across iterations (p-value < 0.01). Among the qualifying input combinations, those input drive pairs for which the circular mean of mean phase differences across iterations lay within 180±22.5 degrees were deemed to be 'in-regime'. For JXK, the gamma-harmonic phase reported in Fig 6 is the circular average of the phase differences from all the iterations. The amplitudes of gamma and its first harmonic in the figure are estimated from the average PSD (examples in Fig 6C and 6D), computed as the mean of the PSDs obtained from 50 iterations for a given stimulus combination.

## Supporting information

**S1 Fig. Mean direction vector strengths of Gamma-harmonic phase differences in JXK vs JS.** (A) Time-averaged direction vector strengths, averaged across 50 iterations of JXK model simulation for the $-r_E$-$r_I$ proxy, computed similarly as for the LFP data. The low vector strengths suggest that the near 180˚ phase differences identified 'in-regime' points were likely not produced by consistent arch-shaped waveforms in subsequent cycles but arose from much rarer sharp troughs in the proxy traces. (B) The mean vector strength computed from the simulation of the JS model for the $-r_E$-$r_I$ proxy. For input combinations that generate oscillations by self-oscillation, subsequent gamma cycles have the same shape, resulting in vector strength of 1. The white contour surrounds the superlinear regime of the JS model as in Fig 7F. The black contours in (A) and (B) encircle the input combinations that satisfied our 'in-regime' criterion on phase differences as in Figs 6F and 7F.
(TIF)

**S2 Fig. Gamma-harmonic phase difference in rate models using different LFP proxies.** (A) Phase difference of gamma and its harmonic in the JXK model computed for each stimulus condition (as in Fig 6F) but using only E population activity to compute the LFP proxy. (B) Gamma-harmonic phase differences in JXK using I population activity alone. (C) Phase differences in JS model using E population activity only. (D) Phase differences in JS using I population activity only. Black contours encircle stimulus conditions identified as 'in-regime'. White contours in (C) and (D) mark the superlinear regime of the JS model.
(TIF)

**S3 Fig. Gamma in JXK model with a different activation function.** JXK model was simulated for an example input combination with a modified activation function obtained by replacing $\lfloor r_I \rfloor$ with $\frac{\lfloor r_I \rfloor^n}{n}$ with n = 2.5 (A-G) Same as Fig 6A–6G (H) Same as S1A Fig.
(TIF)

## Acknowledgments

We thank Dr. Adam Kohn for his helpful comments on the paper.

## Author Contributions

**Conceptualization:** R Krishnakumaran, Supratim Ray.

**Data curation:** Supratim Ray.

**Formal analysis:** R Krishnakumaran, Mohammed Raees, Supratim Ray.

**Funding acquisition:** Supratim Ray.

**Investigation:** R Krishnakumaran, Supratim Ray.

**Methodology:** R Krishnakumaran, Mohammed Raees, Supratim Ray.

**Project administration:** Supratim Ray.

**Resources:** Supratim Ray.

**Software:** R Krishnakumaran, Supratim Ray.

**Supervision:** Supratim Ray.

**Validation:** R Krishnakumaran, Supratim Ray.

**Visualization:** Supratim Ray.

**Writing – original draft:** R Krishnakumaran, Supratim Ray.

**Writing – review & editing:** R Krishnakumaran, Supratim Ray.

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
