## [Decision Letter · Decision Letter 0]

6 Jan 2022

Dear Dr. Ray,

Thank you very much for submitting your manuscript "Shape analysis of gamma rhythm supports a superlinear inhibitory regime in an inhibition-stabilized network" for consideration at PLOS Computational Biology. As with all papers reviewed by the journal, your manuscript was reviewed by members of the editorial board and by several independent reviewers. The reviewers appreciated the attention to an important topic. Based on the reviews, we are likely to accept this manuscript for publication, providing that you modify the manuscript according to the review recommendations.

Sincerely,

Michele Migliore

Associate Editor

PLOS Computational Biology

Thomas Serre

Deputy Editor

PLOS Computational Biology

[LINK]

Reviewer's Responses to Questions

**Comments to the Authors:**

Reviewer #1: Major comments:

— Figures 2 and 5 do not convey the message that wants to be send (or it is difficult to understand). With the aim of improving interpretability, both figures should be rethought to represent the main message better. If, for any reason, the authors want to keep the plots as they are, I suggest to add additional panels with the distribution of values showed in the Y axis of both figures.

— The code in the GitHub repository does not work directly after downloading the folder and follow the steps specified in the Readme. A missing function called mtspectrum is raising an error. If it is part of a mathwork toolbox or has been downloaded from another repository, this should be specified in the Readme, guaranteeing that everyone can reproduce the figures.

Minor comments:

— In line 649 the equation needs to be corrected, the words are badly formatted.

— In line 498: the references to the figures needs to be corrected, there exists a reference to a missing panel.

— I suggest to invert the order of the model parameters in Table 1, such that they match the order they appear in the text and figures, improving consistency and readability.

— Figures 6 and 7 correspond to the same analysis with different models. Consistency between both figures would improve interpretability: please check title style in panels C and D, and box style in panel G.

Reviewer #2: This work prevents a well thought out and relevant comparison of competing models of stimulus-induced gamma activity in comparison to a previously published dataset of gamma activity in macaque. The distinctive oscillation shape, linked to a fixed phase relationship between the gamma and harmonic, is shown to be reproduced sufficiently well in only one of the two models tested.

While the results are already suitable for publication with minor changes, a bit of additional expansion of the analysis and simulations could significantly enhance the impact of this work, completing the picture of the experiment-model match (or lack thereof).

Major comments:

1. In both model and experiment analyses, the authors essentially discard all information about the variability of phase relationships (with the exception of the Rayleigh test to check for nonuniform phase). The vector strength of the circular average of phases could provide a valuable additional comparison (averaged either within or across trials or both). This might also aid in the discussion of whether the stochastic driven model comes close to reproducing the observations – the current explanation that identifies and then dismisses isolated ‘in-regime’ areas in the parameter space feels a bit artificial, although the explanation of the insufficiency of this model makes sense.

2. The authors mention they have some results on models with differently shaped activation functions, but do not show those here. Although fully exploring that space of models would certainly be a full new result in itself, a brief presentation of a variation or two would be very enlightening here, in particular to disentangle the two contrasts of linear/superlinear and persistent/decaying oscillations. A few traces in a supplemental figure, referenced in the relevant discussion in the main text, would be ideal here.

Minor comments:

L48-49: is there a difference between center-frequency and peak-?

L65: “phase does not vary linearly” – with respect to what?

L158: the trial averaging (over the stimulus interval) should be briefly explained here to more clearly distinguish it from the averaging across electrodes

Figure 4: If possible, the phase axis should only span a 180 degree interval to more clearly show the possible domain of phase differences

L197: a brief summary of the structure of the two models would be helpful in the main text here – what are the populations represented and how is the stimulus input represented in the model?

L244: “inhibitory inputs are superlinear” – more accurate to say the inhibitory population response function is in a superlinear regime?

Figure 7A-B: markers are very hard to distinguish, maybe zoom into the parameter space

Figure 8: While this test of alternative LFP proxies is important, I think it could be moved to a supplement to keep the focus on the main results. The repeat of the lack-of-fit of the JXK model here especially adds very little.

L311: Is it possible to interpret the effects of conductances on gamma shape in spiking models as operating through the same shifts in input/response regimes shown in this work?

L389-396: this summary of why the JXK ‘in-regime’ responses are not good fits should move to results, regardless of whether it can be expanded quantitatively using the vector strength analysis mentioned above.

Additional discussion: It would be useful to briefly mention and compare the present superlinear model regime to the supralinear stabilized network regime of (Ahmadian, Rubin and Miller, 2013) – I’m not sure they’re directly comparable, but it seems there may be a contrast where both E and I are superlinear in the SSN vs I only in the JS model?

L659: equation formatting

**Have the authors made all data and (if applicable) computational code underlying the findings in their manuscript fully available?**

Reviewer #1: Yes

Reviewer #2: Yes

PLOS authors have the option to publish the peer review history of their article (what does this mean?). If published, this will include your full peer review and any attached files.

Reviewer #1: No

Reviewer #2: **Yes: **Thomas Chartrand

Figure Files:

Data Requirements:

Reproducibility:

References:

---

## [Editor Report · Decision Letter 1]

31 Jan 2022

Dear Dr. Ray,

We are pleased to inform you that your manuscript 'Shape analysis of gamma rhythm supports a superlinear inhibitory regime in an inhibition-stabilized network' has been provisionally accepted for publication in PLOS Computational Biology.

Best regards,

Michele Migliore

Associate Editor

PLOS Computational Biology

Thomas Serre

Deputy Editor

PLOS Computational Biology

---

## [Editor Report · Acceptance letter]

9 Feb 2022

PCOMPBIOL-D-21-01883R1 

Shape analysis of gamma rhythm supports a superlinear inhibitory regime in an inhibition-stabilized network

Dear Dr Ray,

I am pleased to inform you that your manuscript has been formally accepted for publication in PLOS Computational Biology. Your manuscript is now with our production department and you will be notified of the publication date in due course.

With kind regards,

Olena Szabo
